# Comparison of Two Methods for the Measurement of Blood Plasma and Capillary Blood Glucose in Tropical Highland Grassing Dairy Cows

**DOI:** 10.3390/ani13223536

**Published:** 2023-11-16

**Authors:** Catalina López, Valentina Hincapié, Jorge U. Carmona

**Affiliations:** 1Grupo de Investigación Patología Clínica Veterinaria, Departamento de Salud Animal, Universidad de Caldas, Calle 65 No 26-10, Manizales 170004, Colombia; catalina.lopez@ucaldas.edu.co (C.L.); valentina.531714405@ucaldas.edu.co (V.H.); 2Grupo de Investigación Terapia Regenerativa, Departamento de Salud Animal, Universidad de Caldas, Calle 65 No 26-10, Manizales 170004, Colombia

**Keywords:** bovine, glucometer validation, energy metabolism

## Abstract

**Simple Summary:**

Blood glucose is crucial for milk production in dairy cows. The use of some human-customized blood glucometers has been evaluated for the measurement of cow blood with acceptable outcomes. However, there is scarce information validating the performance of cow-customized blood glucometers, like Centrivet GK (CVGK). We evaluated and compared the performance of the glucometer CVGK in tropical highland grazing cows against the measurement of glucose concentrations in plasma and serum via an enzymatic/photometric assay, considering the glucose measurements in serum as the reference method. Our study indicates that the measurement of glucose concentrations in plasma or by using CVGK is not reliable compared to the reference method used in our research. Thus, in the environmental and technical conditions of the study, the use of this cow-side glucometer cannot be recommended for glucose measurement in cows.

**Abstract:**

(1) Background: There is lack of published studies validating specific cow-side glucometers such as Centrivet GK (CVGK). (2) Methods: The aims were (1) to measure and compare the blood glucose concentrations in 52 tropic highland grassing cows by using CVGK and the traditional enzymatic/photometric assay (EPA) in plasma and serum (reference method) and (2) to establish if glucose concentrations obtained via these methods could be affected by several demographic and zootechnical parameters of the dairy herd evaluated. (3) Results: Glucose concentrations were significantly (*p* = 0.00) affected by the method used for their measurement. The intra-assay coefficient of variation (CV) for glucose concentrations in plasma EPA and for CVGK was 14% for both methods with serum EPA, whereas the inter-assay CV for plasma EPA and CVGK was 8% and 13.7%, respectively, with serum EPA. Pearson correlation coefficient calculations between the reference method in serum and plasma presented a slightly positive significant (*p* = <0.000) correlation (r = 0.56), whereas there was not a significant (*p* = 0.413) correlation between serum EPA and CVGK (r = 0.135). The Passing and Bablok regressions were out of the ideal expected values for the slope (β = 1) and the intercept (α = 0) (11), whereas the Bland–Altman plots showed a bias of 5.29 ± 11.73 (mg/dL) for serum and plasma and 11.01 ± 15.74 (mg/dL) for serum and CVGK. The ROC curve showed no sensitivity in detecting normoglycemic cows (area = 53.7 %, e.d = 12.5 %, *p* = 0.759) for CVGK when compared to plasma EPA (area = 36.1 %, e.d = 14.2 %, *p* = 0.256). Plasma EPA exhibited a better but not significant effect in detecting hyperglycemic cows (area = 63.9%, e.d = 14.2%, *p* = 0.256) when compared to HHD (area = 46.3 %, e.d = 12.5 %, *p* = 0.759). General glucose concentrations, independently of the method used, were significantly (*p* = <0.001) greater in young cows when compared to adult and old cows. (4) Conclusions: Glucose concentration measurement in plasma by using EPA or in capillary blood via CVGK were not reliable methods when compared with the reference method.

## 1. Introduction

Glucose is a critical metabolite for dairy cows during pregnancy and lactation [1,2,3,4]. This substance is highly required at the beginning of lactation; hence, its diminution in blood puts parturient cows at risk of negative energy balance and the subsequent development of ketosis [5,6]. Glucose is pivotal in those groups of cows that are milked and are pregnant. Lower glucose concentrations could affect ovarian function and diminish the phagocytic activity of leukocytes in the reproductive tract of recent parturient cows, which could produce uterine infection and diminish conception after the first postpartum artificial insemination [6,7].

Lactating and end-term pregnant cows are frequently affected by ketosis as a consequence of insulin resistance [8,9]. However, this pathophysiological mechanism is considered a metabolic adaptation to maintain the energy supply provided by glucose to ensure the fetal vital functions and the future nourishment of the calf at the moment of calving [7]. From a biochemistry point of view, bovine ketosis is a chronic clinical state induced by increased plasma concentrations of β-hydroxybutyric acid (BHBA) [9].

Two clinicopathological forms of bovine ketosis have been described. Type I ketosis is characterized by a hypoglycemic/hypoinsulinemic state, which is manifested 3–6 weeks after calving in high-milk production cows, whereas type II ketosis displays higher concentrations of glucose and insulin and is frequently associated with overfeed and fattened cows during the dry period [2]. Furthermore, this disease can be classified in subclinical and clinical forms. The first form presents BHBA blood concentrations ranging between 1.2 and 1.4 mmol/L in apparently healthy cows over the first 14 days of lactation. On the other hand, cows with clinical ketosis usually exhibit BHBA concentrations higher than 1.4 mmol/L [5].

Routine glucose and BHBA blood measurements are of paramount importance in dairy farms to obtain an early diagnosis of ketosis and to establish both adequate therapy and a nutritional program to avoid health and production problems in cows [5,7]. There are several methods to determine the concentration of these mediators in blood, which involve the restriction and venipuncture of the animals [2,5]. This not only represents a risk for the animal and farm personnel but also increases the probability of obtaining altered samples with higher glucose concentrations induced by stress [2]. In line with this, several hand-held devices have been proposed for glucose and BHBA cow-side determination [5,6,9,10]. However, there are scarce published studies in which the measurement of glucose concentrations is compared with bovine-specific hand-held devices and traditional biochemical tests by using either dry or anticoagulant-charged tubes [2].

The aims of the present study were (1) to validate the use of a hand-held commercial device (Centrivet GK, ACON Laboratories Inc., San Diego, CA, USA) calibrated for bovine blood to measure the capillary blood glucose concentrations in tropic highland grassing cows in comparison to the glucose measurement in plasma (obtained in tubes with sodium fluoride/potassium oxalate (15 mg/12 mg) as an anticoagulant) and serum by using an enzymatic/photometric assay (EPA), considering this last technique as the reference method, and (2) establish if glucose concentrations obtained via the evaluated methods could be affected by several demographic and zootechnical parameters of the dairy herd evaluated in the study, such as age group, breed, reproductive status, pregnancy status, and days of lactation.

## 2. Materials and Methods

This animal study protocol was approved by the Committee for Animal Experimentation of the Universidad de Caldas ((UC-0616617783-22), date of approval: 6 June 2022). The dairy herd of the study is owned by the Universidad de Caldas, Manizales, Colombia, who authorized the sampling of the cows. The experiments were conducted according to Colombian animal welfare policies.

### 2.1. Farm Location and Animals

The study was conducted in a dairy farm located at 2352 m.a.s.l, at 5°01′46″ N 75°25′43″ W, in the coffee region of Colombia. Fifty-two dairy cows of several ages and breeds, with different milk production and reproductive statuses, were included. Cows were managed under rotational grazing systems and received supplementation with balanced food according to milk yield. The predominant pasture was Kikuyu grass (*Pennisetum clandestinum*). The concentrates used were commercial mixes of cereals, containing between 14 and 16% of crude protein, and approximately 2.9 Mcal ME/kg of dry matter. The concentrates were fed starting three weeks before calving (2 kg/cow/day) and at a rate of 1 kg per 4 kg of milk yield after calving. Mineral supplements and water were available ad libitum. The cows were milked twice a day.

### 2.2. Design of the Study

All the cows were sampled twice to obtain venous and capillary blood, before (05:30 h) and after (07:00 h) milking. Of note, the animals were mainly grassed until milking took place, and only food was supplied during the milking time.

### 2.3. Blood Sampling and Processing

After animal restriction, three blood samples were obtained. One sample (blood drop) was obtained by using a lancet puncture in the lateral side of the vulvar region after cleaning and disinfection of the skin area. This sample was used for glucose and BHBA determination by using a cow-specific hand-held device (Centrivet GK, ACON Laboratories Inc., San Diego, CA, USA). Two additional 6 mL whole-blood samples were obtained from the coccygeal vein and deposited in both a clot activator plain tube (KJ0601Z, Pro-Coagulante, Kangjian Medical, Jiangsu Province, Thaizhou, China) and in a tube with sodium fluoride/potassium oxalate (15 mg/12 mg) (BD Vacutainer, ref 367925, Becton Drive, NJ, USA). After collection, blood tubes were maintained at 4 °C and centrifuged over the first hour at 3000 g/6 min. Then, glucose concentrations were measured (in either serum or plasma) via an enzymatic/photometric assay (BTS350 Chemistry Analyzer, BioSystems, Barcelona, Spain).

All the glucose measurements were performed in triplicate for each cow per each method. A summary of the design of the study is presented in Figure 1.

### 2.4. Statistical Analysis

The power of the present study was calculated by considering a β value of 0.8 and an α value of 95%. Data generated were explored for normality by using the Kolmogorov–Smirnov test.

In general, BHBA and glucose concentrations (obtained for all methods) were presented by descriptive statistics. Initially, BHBA concentrations in healthy and ketosis cows were evaluated using a generalized linear mixed model (GLMM).

Glucose concentrations obtained via the three methods were firstly assessed using several statistical techniques that which the Pearson correlation coefficient calculations, Passing and Bablok regression, and the generation of Bland and Altman plots and calculations. In general, these techniques were carried out to establish the potential concordance between the evaluated methods for glucose concentrations considering the enzymatic/photometric assay (EPA) performed in serum as the reference method. A GLMM was performed to establish if glucose concentrations were significantly affected by the three methods used for their measurement at two times (including the evaluation of the interaction of these factors).

Additional GLMMs were performed to establish how glucose concentrations could be affected by the method used for their measurement in combination with some demographic and zootechnical factors of the dairy herd studied, such as the age group (young cows (under 60 months of age), adult cows (animals between 61 months and 96 months of age), and old cows (animals older than 97 months)), breed (Holstein, Normande, F1, crossbreed, and other cows), reproductive status (empty, pregnant, and inseminated cow), pregnancy status (non-pregnant, early-pregnant (day 1 to 95), middle-pregnant (day 96 to 190), and late-pregnant (day 191–285)), and days of lactation (up to 60 days of lactation; 61 to 120 days of lactation; over 120 days of lactation). This additional statistical analysis was performed to identify incidental findings related to the zootechnical farm profile and how glucose concentrations could be affected in the cows enrolled in our study.

According to the classification for glucose concentration proposed by Mair et al. (2016), cows were classified as hypoglycemic (<40 mg/dL), normoglycemic (40–60 mg/dL), and hyperglycemic (>60 mg/dL), and then, sensitivities (Se) and specificities (Sp) for the plasma enzymatic/photometric assay and hand-held device (glucometer) were obtained, considering the serum glucose determinations as the reference method. Then, a receiver operating characteristics (ROC) analysis was performed to establish the ideal thresholds to identify the samples according to the aforementioned glucose concentration’s classification.

In all the GLMMs performed in this study, the interaction between the factors was also analyzed, and the cow ID was declared as a random factor. In general, when models exhibited significant differences, a Tukey test was performed. For all the GLMMs performed, Gaussian distributions were employed, and the identity was set as a link function. A *p* < 0.05 was accepted as significant for all the tests performed.

Data were assessed by using three statistical software: JASP 0.18.1 (University of Amsterdam, The Netherlands), Real Statistics Resource Pack software ((Release 8.8.1). Copyright (2013–2023) Charles Zaiontz. www.real-statistics.com, accessed on 1 September 2023), and SPSS 25 (IBM, New York, NY, USA). JASP was used for performing all GLMMs and Spearman correlation tests, Real Statistics Resource Pack software was used for calculating the Bland–Altman and Passing–Bablok regression values, and SPSS 25 was employed for ROC calculation and for generating ROC curve plots.

## 3. Results

A total of 52 cows were included in the study with a mean age of 62.52 (±30.38) months (range 26.30–143.4 months) and a mean milk yield of 19.79 (±8.31) L/d (range 0–38.80 L/d). Regarding the age group of the animals, 31 (59.62%) were young, 13 (25%) were adult, and 8 (15.38%) were old cows. Regarding the classification of days of lactation, 9 (15.38%) cows had up to 60 days of lactation, 10 (19.23%) animals had between 61 and 120 days of lactation, and 33 (63.46%) cows had over 120 days of lactation.

This group of animals was composed of 23 (44.23%) Holstein, 10 (19.23) Normande, 13 (25%) crossbreed, and 6 (11.54%) F1 (Holstein/Gyr) cows. Regarding the reproductive status of the animals, the group was composed of 29 (55.77%) pregnant, 14 (26.92%) empty, and 9 (17.31) inseminated cows, whereas regarding the pregnancy status of the animals, 23 (44.23%) were non-pregnant, 11 (21.15%) were early pregnant, 13 (25%) were middle-pregnant, and 5 (9.62%) were late-pregnant cows. Of the 52 cows included, 48 (92.31%) animals presented BHBA concentrations < 1.2 mmol/L (healthy cows), whereas 4 (7.69%) cows had BHBA concentrations > 1.2 mmol/L (ketosis cows) at the two times of evaluation, before and after milking (Figure 2A).

Glucose concentrations were significantly (*p =* 0.00) affected by the method used for their measurement. However, there were no significant effects related to the time or the interaction between these two factors (Table 1 and Figure 2B). Bearing in mind that glucose concentrations were not affected by time and by the interaction between method and time, data from the two measurement times for each method were unified to perform Pearson correlation coefficient calculations, Passing and Bablok regression, and the generation of Bland–Altman plots and calculations. The intra-assay coefficient of variation (CV) for glucose concentrations measured in plasma via EPA and for HHD was 14% for both methods regarding serum EPA, whereas the inter-assay CV for serum EPA and HHD was 8% and 13.7%, respectively, in comparison to serum EPA.

Pearson correlation coefficient calculations between the reference method used for measuring glucose concentrations in serum and plasma presented a slightly positive significant (*p =* <0.000) correlation (r = 0.56) (Figure 3A), whereas there was not a significant (*p* = 0.413) correlation between the glucose measurement in serum and capillary blood measured using a hand-held device (r = 0.135) (Figure 3B). Passing and Bablok regression and Bland–Altman calculations values are presented in Table 2. In general, results from Passing and Bablok regression for comparisons between glucose measurements in serum and plasma and serum and the hand-held device (capillary blood) were out of the ideal expected values for the slope (β = 1) and the intercept (α = 0) [11].

In this sense, there was no agreement between the glucose concentrations measured in plasma or by using the hand-held device in serum. On the other hand, the Bland–Altman plots showed a bias of 5.29 ± 11.73 (mg/dL) (Figure 4A) for comparisons between serum and plasma and of 11.01 ± 15.74 (mg/dL) for serum and the hand-held device (Figure 4B).

Regarding ROC, curve analysis we found that no cow was hypoglycemic according to the previous classification used in this study. According to the results of 104 samples taken using the EPA in serum, 6 samples (5.8%) were classified as normoglycemic and the rest of the samples (98 samples (94.2%) were classified as hyperglycemic. Thus, the ROC analysis was only performed for normo- and hyperglycemic cows. In this case, the HHD for glucose determination in capillary blood showed apparent higher sensitivity in detecting normoglycemic cows (Area = 53.7%, e.d = 12.5%, *p =* 0.759) when compared to the measurement of the same analyte in plasma via EPA (Area = 36.1%, e.d = 14.2%, *p =* 0.256). Conversely, the plasma EPA method exhibited a better but not significant effect in detecting hyperglycemic cows (area = 63.9%, e.d = 14.2%, *p =* 0.256) when compared to HHD (area = 46.3%, e.d = 12.5%, *p =* 0.759) (Figure 5A,B).

GLMM results describing how the glucose concentrations could be affected by the method used for their determination and by several demographic and zootechnical parameters are presented in Table 3. In general, after performing five GLMMs, the “Method” factor always presented significant effects (*p* = <0.001) on the models evaluated (Table 3).

Furthermore, we observed that the “age group” factor significantly (*p* = <0.001) influenced model 1, but this effect was not related to the interaction with the method factor (Table 3). In line with this, general glucose concentrations, independently of the method used, were significantly (*p* = <0.001) higher in young cows, when compared to adult and old cows (Figure 6A,B).

## 4. Discussion

Glucose concentration monitoring is an essential tool for preventive medicine programs and diagnostic aid in bovine practice [7]. Patient-side glucometers have allowed for an important advancement in the field management of several dairy-herd-related diseases, such as ketosis [5,6], insulin resistance [12], and neonatal calf diseases [13], amongst others. Although several research works that have evaluated the use of human-calibrated glucometers for measuring glucose concentrations in cows have been described [2,12,13,14,15,16], according to our literature search, we found scarce published studies about the use of specie-specific glucometers for bovines [17,18]. Interestingly, in these two studies [17,18], the bovine glucometer Centrivet GK (ACON Laboratories Inc., San Diego, CA, USA) was used for measuring glucose concentrations; however, it was not possible to find peer-reviewed publications about the validation of this hand-held device for bovine glucose measurement in field or experimental conditions against a reference method for glucose measurement.

For comparative proposals, we defined the glucose determinations in serum via EPA as the reference method [14] instead of using the valuation of this metabolite in plasma via the same technique, because significantly higher concentrations of glucose were obtained in serum samples when compared to plasma samples.

Authors like Wittrock et al. [15] established as a reference method the measurement of glucose concentration in plasma according to the conception that sodium fluoride/potassium oxalate preservatives could stop glucose metabolism in blood cells by disrupting the glycolytic cascade [19]; however, as mentioned, our results were contrary to those established by these authors [15,19].

We decided to perform the present pilot study in the typic environmental and social conditions of technical highland tropical dairy farms of our country, which is represented by a small number of animals in milking and with health and zootechnical parameters similar to those presented in the results of this manuscript to mainly validate these kind of cow-side glucometers in the influence area of our educative institution to develop potential research and extension programs in local dairy herds by using a validated bovine glucometer.

In the present study, three different statistical methods (Pearson coefficient calculations, Passing and Bablok regressions [11], and Bland–Altman plots and calculations [20]) were used to establish the degree of concordance of the glucose measurements between the reference method (serum EPA) and those measured in plasma via EPA and using the Centrivet GK glucometer. In general, our results derived from these statistical tests showed a very low level of concordance for glucose concentrations measured between the reference method and plasma via EPA and no concordance between the reference method and the glucometer. In general, glucose concentrations measured using the Centrivet GK device were 13.7% lower than the concentrations of the same metabolite measured in serum. Regarding this finding, Katsoulos et al. [14] experienced a similar situation and hence hypothesized several causes for this phenomenon such as low blood impregnation of the device stripes and high hematocrit values. However, the experiments were conducted rigorously, and the hematocrit levels of the cows were always between the ranges suggested by the glucometer’s manufacturer.

As reported by Mair et al. [2], no hypoglycemic cows were detected in our study, despite the dairy herd evaluated having some cows at early lactation with expected negative energy balances. These cows may have displayed a physiological insulin resistance state, such as that previously hypothesized by Holtenius and Holtenius [21]. However, it seems to be that glucose is an insensitive indicator of energy status in cattle due to its homeostatic control [22,23,24,25]. In line with this, additional research is necessary to establish cut-off values for glycemia status classification in our environmental conditions [22,26]. At this point, it is necessary to consider that we performed a ROC analysis to determine the capacity of the evaluated methods for detecting normo- and hyperglycemic cows. Although the glucose measurements in plasma via EPA or via the cow-side glucometer presented some sensitivity and specificity to detect these clinical groups of cows, the level of significance was not acceptable; hence, both methods were not demonstrated to be useful for discriminating normo- and hyperglycemic cows compared to the reference test [2].

In our study, an intriguing finding was related to the fact that young cows (independently of the method used for glucose measurement) presented significantly higher concentrations of this metabolite in comparison to adult and old cows, independently of the reproductive status of the studied animals. To note, several studies did not report an effect of age on glucose concentrations in dairy cows [3,16]. However, several studies have shown that glucose concentrations could be significantly affected by the pregnancy status of the cows [26,27,28,29]. It is possible that young cows grazing in high-altitude tropical conditions have a greater intake of pastures than adult and old cows or present a different metabolic energy profile that makes these animals present higher glucose concentrations in comparison with other age groups [30,31,32,33,34]. However, this is only a hypothesis that should be proved through controlled studies.

## 5. Conclusions

Glucose concentration measurement by using the glucometer Centrivet GK in capillary blood or in plasma by using EPA in the cows of the study were not reliable methods when compared with the reference method used in the study; thus, further studies in a major number of animals are necessary to validate or reject the results of the present study.

On the other hand, we incidentally found that young cows in our study exhibited higher concentrations of blood glucose than adult and old cows, independently of the method used for glucose measurement or of their reproductive status. The zootechnical or clinical importance of this finding remains to be clarified.

## Figures and Tables

**Figure 1 animals-13-03536-f001:**
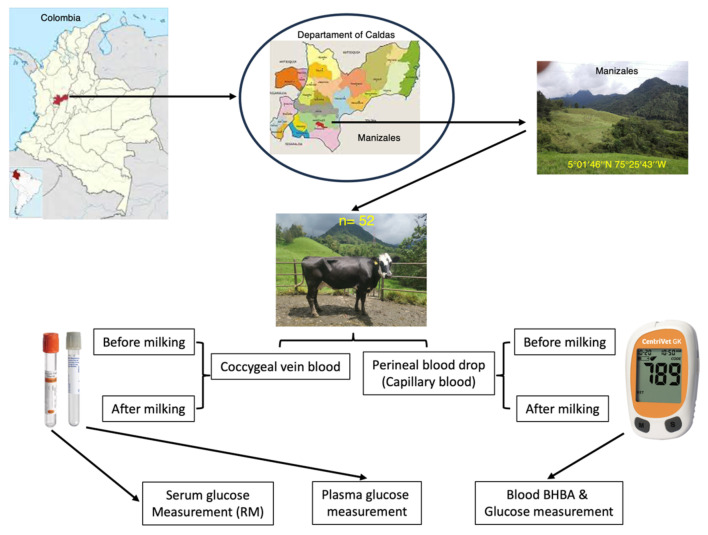
Design of the study. BHBA = beta hydroxybutyric acid, RM = reference method.

**Figure 2 animals-13-03536-f002:**
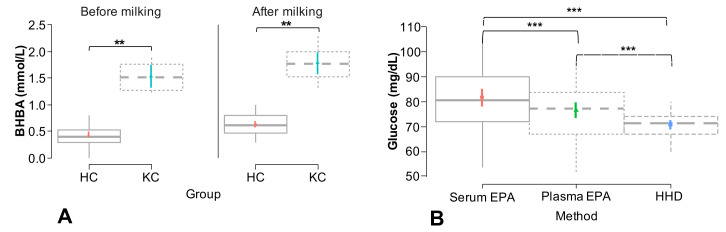
(**A**) Mean β-hydroxybutyric acid concentrations (mmol/L) in healthy cows (HC) and ketosis cows (KC) before and after milking. (**B**) Total mean glucose concentrations (mg/dL) obtained in serum and plasma using an enzymatic/photometric assay (EPA) and in capillary blood using a hand-held device (HHD). ** = indicate significant differences (*p =* <0.05) between the evaluated groups. *** = indicate significant differences (*p =* <0.001) between the evaluated groups.

**Figure 3 animals-13-03536-f003:**
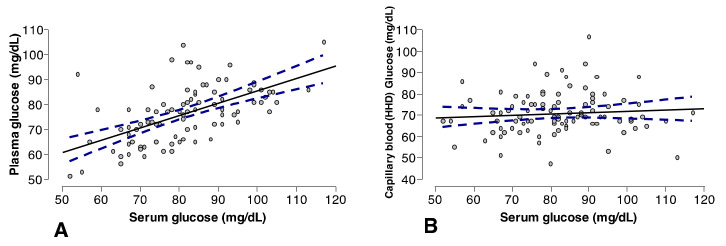
Pearson correlation plots for (**A**) serum glucose measured via EPA versus plasma glucose measured via EPA (r = 0.56, *p =* <0.000) and (**B**) serum glucose measured via EPA versus capillary blood measured via HHD (Centrivet GK) (r = 0.135, *p* = 0.413).

**Figure 4 animals-13-03536-f004:**
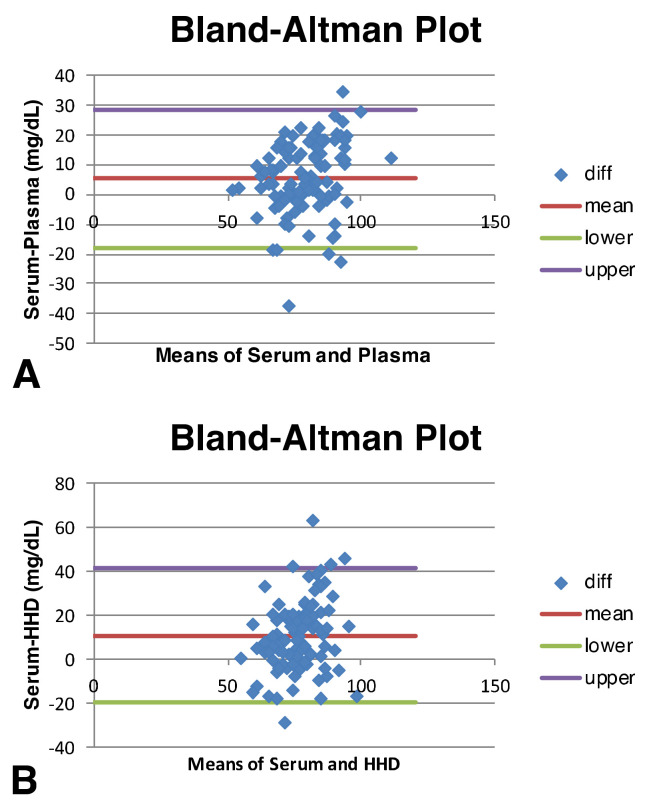
Bland–Altman plots of mean differences in glucose concentrations measured in (**A**) plasma via EPA and (**B**) capillary blood measured via HHD (Centrivet GK) against the reference method. The red line represents the mean; the solid upper and lower lines represent the mean ± 1.96 SD.

**Figure 5 animals-13-03536-f005:**
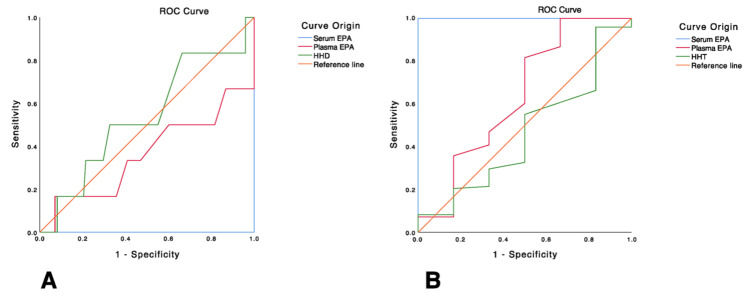
Receiver operating characteristics (ROC) curves for (**A**) detecting normoglycemia and (**B**) detecting hyperglycemia considering the determination of glucose concentration in serum using the enzymatic/photometric assay (EPA) as the reference method. HHD = hand-held device.

**Figure 6 animals-13-03536-f006:**
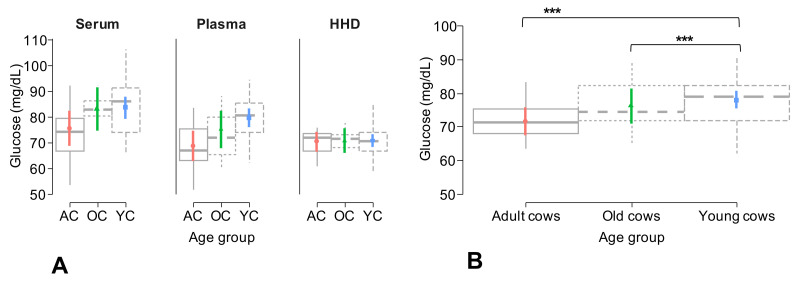
(**A**) Generalized linear mixed model (GLMM) plot showing the effect of the interaction of the method used for glucose determination and the age group of the cows of the study. (**B**) GLMM plot describing the individual effect of the age group on glucose concentration independently of the method used for its determination. *** = indicate significant differences (*p* = <0.001) between the evaluated groups.

**Table 1 animals-13-03536-t001:** General descriptive statistics for the obtained glucose concentrations (mg/mL) via the three methods in 52 cows.

Statistical Parameter	Method Used for Glucose Measurement (mg/mL)		
	Serum EPA (Reference Method)	Plasma EPA	Hand-Held Device (Capillary Blood)
Before milking			
Mean	81.07 ^a^	78.92 ^b^	71.25 ^c^
95% IC mean upper	84.35	82.40	73.64
95% IC mean lower	77.98	75.44	68.85
Standard deviation	11.73	11.78	15.10
Median	81.00	78.00	70.00
Interquartile range	55.00–113.00	51.00–105.00	50.00–91.00
After milking			
Mean	82.00 ^a^	73.72 ^b^	69.83 ^c^
95% IC mean upper	86.19	86.19	72.73
95% IC mean lower	77.81	77.80	66.91
Standard deviation	15.11	10.69	10.71
Median	81.50	72.50	67.00
Interquartile range	52.00–117.00	53.00	47.00–107.00

^a–c^ = showcase letters denote significant (*p* < 0.001) differences obtained via the Tukey test for glucose concentrations obtained using the different evaluated methods. Note that the time factor did not affect the model.

**Table 2 animals-13-03536-t002:** Evaluated glucose concentration (mg/dL) differences via two different methods using serum determination with the enzymatic/photometric assay (EPA) as a reference test by employing the Passing–Bablok regression and Bland–Altman analysis methods.

Method for Glucose Determination		Statistical Method							
	Passing–Bablok Regression Values					Bland–Altman Calculations Values			
	Slope (β)	CI95% (β)	Intercept (α)	CI95% (α)	*p* Value	Mean Diff	SD Diff	Lower	Upper
Plasma (EPA)	0.72	0.61–0.92	14.81	0.27–26.6	0.09	5.27	11.73	−17.73	28.28
Hand-held device (capillary blood)	2.00	1.33–3.20	−57.00	3.20–138.8	0.01	11.01	15.74	−19.87	41.87

**Table 3 animals-13-03536-t003:** GLMMs evaluating the effects of the method used for glucose assessment and how several demographic and zootechnical parameters in the cows of the study can affect the concentrations of this analyte.

GLMM Number	Effect	*p*-Value
1	Intercept	<0.001
	Method	<0.001
	Age group	0.035
	Method × Age group	0.075
2	Intercept	<0.001
	Method	<0.001
	Breed	0.752
	Method × Breed	0.913
3	Intercept	<0.001
	Method	<0.001
	Reproductive status	0.696
	Method × Reproductive status	0.610
4	Intercept	<0.001
	Method	<0.001
	Pregnancy status	0.679
	Method × Pregnancy status	0.697
5	Intercept	<0.001
	Method	<0.001
	Days of lactation	0.940
	Method × Days of lactation	0.533

## Data Availability

Research data from this manuscript can be requested from J.U.C. (carmona@ucaldas.edu.co).

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
