# Peer review of "Comparison of Two Methods for the Measurement of Blood Plasma and Capillary Blood Glucose in Tropical Highland Grassing Dairy Cows"

_animals, 2023, doi:10.3390/ani13223536_

Round 1

Reviewer 1 Report

Review the subtopics of the summary.

Reduce the size of the summary.

Keywords: Don't use the words in the title.

Line 106: Include the Ethics Committee authorization number.

Line 107: Write down the institution where the experiment took place.

Lines 164-181: This information should not be included in the statistical analysis. Review the best way to present this information.

More objective in statistical analysis.

Table 1 is in the wrong place, put it inside the results.

Lines 221-226: Proofread the text and set the units internationally. The p-value is incomplete.

Line 228: What is the value of p?

Proofread the unit of measurement for the entire manuscript. They're not international.

Figure 4: The diagram is represented according to Figure 3.

Line 333: What is the value of p?

I recommend accepting the article, however it is necessary to review the English and correct some identified points.

Author Response

Rebuttal letter for Reviewer 1.

Dear Reviewer 1:

Thank you very much for your time and effort in reviewing our manuscript and helping us to improve the new version of the document. Regarding your concerns, we addressed the same as follow:

  1. The subtopics of summary and abstract were revised and the size of both paragraphs was reduced. Please, see line (L) 12-44. Furthermore, keywords were modified (L 45).
  2. Regarding your concerns about the ethical statement of the manuscript, we performed changes in the new document accordingly. Please, see L 97-100.
  3. We made some modifications in the statistical analysis section according to your suggestions. Please, see L 156-164.
  4. According to your suggestion, Table 1 was put in the section of results. Please, see L 192-197.
  5. The manuscript was entirely revised for p values and correct units of measurement. Some examples appear in L 216, 324.
  6. Figures 3 and 4 were generated with different data.
  7. The new version of the manuscript was entirely revised for English proofreading.

Reviewer 2 Report

animals-2639470 - Comparison of two methods for the measurement of blood plasma and capillary blood glucose in tropical highland grassing dairy cows

1)  The aims of this study were 1) to measure and compare the blood  glucose concentrations in 52 tropic highland grassing cows at different stages of milk production by using CVGK and traditional enzymatic/photometric assay (EPA) of this analyte in plasma and serum (reference method) and, 2) establish if glucose concentrations obtained by the evaluated methods could be affected several demographic and zootechnical parameters. The main contribution is that bovine specific Centrivet GK was not a reliable method to determine glucose.  There were sufficient numbers of animals and at different stages of production to have a good evaluation of glucose.

2) Overall this was a well written paper and easy to follow.  There was a testable hypothesis.  There were a few grammatical mistakes throughout the manuscript, but it does add to the literature on how samples should be collected and handled to accurately determine glucose concentrations.

Throughout the manuscript replace stablish with establish

In the description of the age of cows how can both groups be adults?  And what does (15,38%) stand for?

Table 2 – Add in the serum treatment.

3)

Page    Line                 Comment

1          14                    replace probed with evaluated

1          15                    Delete “for some of these hand-held devices

1          16                    insert is to read there is scarce

1          20                    replace shows with indicates

1          31                    insert by to read affected by

2          48                    replace higher with greater (higher refers to an altitude)

2          58                    delete the

2          69                    delete the

2          89                    replace be with are

3          127                  move was to read food was only supplied

3          139                  should be 4°C

5          191                  delete the

8          298                  change to several research works have evaluated the

Throughout the manuscript replace stablish with establish

Author Response

Rebuttal letter for Reviewer 2

Dear Reviewer 2:

Thank you very much for your time and effort in reviewing our manuscript and helping us to improve the new version of the document. Regarding your concerns, we addressed the same as follow:

  1. The new version of the manuscript was entirely revised for English proofreading.
  2. The word stablish was replaced by established throughout the manuscript. For some examples, please, see lines (L) 26, 72, 91 and so for.
  3. The age group classification of the cows in the section of results was corrected. Please, see L198.
  4. Regarding your suggestion to add information about serum treatment in table 2, it is important to consider that the information presented for Plasma EPA and Centrivet GK was obtained by using data from serum EPA as a reference method as mentioned in table 2 paragraph.
  5. Finally, all your grammar suggestions were considered. Please, see L 13, 14, 18, 41 and so for.

Reviewer 3 Report

Laboratory blood tests are performed to diagnose the disease, detect subclinical conditions in clinically healthy individuals, determine the system affected by the pathological process and the degree of its damage, make a diagnosis and prognosis, or to monitor the treatment process. Performing veterinary laboratory tests most often includes blood tests, which are the most common additional tests performed on patients in veterinary offices and clinics and, as veterinarians emphasize, are necessary in the diagnosis of many diseases.To speed up analyzes and the process of obtaining the results, veterinary facilities often decide to purchase professional equipment and perform the tests themselves. This may become one of the elements of gaining a competitive advantage for a veterinary clinic in terms of product, that is the scope of provided services. Basic laboratory tests are often the standard, unfortunately this does not correspond with the quality of the performed tests.

There are no general standardized guidelines for the storage and transport of blood samples. In practice, test results obtained in different laboratories and determined using different methods are subjected to errors that are difficult to estimate.There are many biochemical parameters that are stable over time, but equally many that change from the moment the patient is prepared to collect blood. The changes are influenced by the skills ability to collect the sample and its stability depending on external conditions, but above all, accuracy and precision of the reference method procedure. It turned out to be a mistake to perform laboratory tests on animals using equipment dedicated to humans. The calibration of such devices differs due to the reference values of different species for the same parameter, therefore the experiment proposed by the authors contains an error in the assumption of the research hypothesis: ”comparison of the glucose concentrations obtained by the mentioned methods with the measurement of the same analyzer in serum by using a EPA, which is considered as the reference method, and determining if glucose concentrations obtained by the evaluated methods could be affected several demographic and zootechnical parameters of the dairy herd evaluated in the study.” This has long been confirmed. In addition, it is compounded by pre-laboratory errors which value is estimated at up to 46-68% in the overall research process.

Considering the value of the results obtained by the authors, I propose to change the assumption (research hypothesis) and use it to validate the bovine glucometer Centrivet GK (ACON Laboratories Inc, San Diego, CA, USA), for which no validation was found in the literature.

Author Response

Rebuttal letter for Reviewer 3.

Dear Reviewer 3:

Thank you very much for your time and effort in reviewing our manuscript and helping us to improve the new version of the document. We enjoyed your valuable commentaries about the general picture of the use of simple diagnostic aids to replace those that are used as standard laboratory methods and the challenges that could implicate these paradigmatic changes.

On the other hand, according to your suggestion, we modified the aim 1 of our research work. Please, see L 81-89 in the new version of the manuscript.

Round 2

Reviewer 3 Report

I appreciate the immediate reaction to my commentaries. However, I insist that the Authors should introduce certain changes so that the paper is a complete whole. In reference to the hypotheses presented in the study (92-102), the conclusion is not sufficiently developed, especially since the integrals obtained by this method cannot be expressed in analytical form. My suggestion is that the Authors should expand the conclusions towards a more analytical presentation because the effect of demographic and zootechnical factors on the level of glucose has been confirmed before and, therefore, it should be revealed in all the methods used in the study.

Author Response

Rebuttal letter for Reviewer 3 (R2).

Dear Reviewer 3:

Once more, thank you very much for your time and effort in reviewing the version 2 of our manuscript, in line with your comments:

I appreciate the immediate reaction to my commentaries. However, I insist that the Authors should introduce certain changes so that the paper is a complete whole. In reference to the hypotheses presented in the study (92-102), the conclusion is not sufficiently developed, especially since the integrals obtained by this method cannot be expressed in analytical form. My suggestion is that the Authors should expand the conclusions towards a more analytical presentation because the effect of demographic and zootechnical factors on the level of glucose has been confirmed before and, therefore, it should be revealed in all the methods used in the study.”

We made some changes in the manuscript including a small change of the conclusion (Please, see lines (L) 383-386. Furthermore, in the M & M section we added a paragraph to explain the reason why this part of the statistical analysis was performed (Please, see L 164-167). Finally, we made a new literature search and only found a study (1) in which the glucose concentration of studied cows was significantly affected by the reproductive status, but we were unable to find additional research works describing similar results to those obtained in our study. This new reference was included and discussed in the new version of the manuscript (Please, see L368-370, and 459-460).

We hope to have solve all your concerns and are open to new suggestions that can improve our manuscript. Best regards,

The Authors.

Reference

  1. Mohammed SE, Ahmad FO, Frah EAM, Elfaki I. Determination of Blood Glucose, Total Protein, Certain Minerals, and Triiodothyronine during Late Pregnancy and Postpartum Periods in Crossbred Dairy Cows. Vet Med Int. 2021;2021:6610362.